# Cluster-Based JRPCA Algorithm for Wi-Fi Fingerprint Localization

**Li Zhang** [1,*] , **Min Zhang** [1] , **Jingao Xu** [2] **and Yi Xu** [1]

1   School of Mathematics, Hefei University of Technology, Hefei 230009, China
2   School of Software, Tsinghua University, Beijing 100084, China
*   Correspondence: lizhang@hfut.edu.cn

**Abstract:** Indoor localization services are emerging as an important application of the Internet of Things, which drives the development of related technologies in indoor scenarios. In recent years, various localization algorithms for different indoor scenarios have been proposed. The indoor localization algorithm based on fingerprints has attracted much attention due to its good performance without extra hardware devices. However, the occurrence of fingerprint mismatching caused by the complexity and variability of indoor scenarios is unneglectable, which degrades localization accuracy. In this article, by combining weighted kernel norm and $L_{2,1}$-norm, a joint-norm robust principal component analysis (JRPCA in brief) assisted indoor localization algorithm is proposed, which can improve the localization accuracy through aggregating the reference points (RPs) and conducting robust feature extraction based on clustering. More specifically, a one-way hierarchical clustering termination method is proposed to obtain reasonable RP clusters adaptively according to the preset RPs. A two-phase fingerprint matching algorithm of JRPCA based on clustering is proposed to further increase the difference between similar RPs, thus facilitating rapid identification and reinforcing localization accuracy. To validate the proposed algorithm, extensive experiments are conducted in real indoor scenarios. The experimental results confirm that the proposed cluster-based JRPCA algorithm outperforms other existing algorithms in terms of robustness and accuracy.

**Keywords:** weighted kernel norm; robust PCA; clustering; fingerprints; RSSI





## 1. Introduction

The Global Positioning System (GPS) [1] has been successfully applied in outdoor environments with high positioning accuracy in practice. In general, GPS technology mainly relies on the propagation of signals through the air. However, when encountering complex buildings (such as supermarkets, commercial centers, hospitals, airports, etc.), signal transmission is easily subjected to interference from plenty of uncertain factors. Weak reception of signals, lack of line of sight between users and satellite, radio multi-path effects, as well as dispersal and fading in indoor environments all contribute to the poor indoor performance of GPS. As a result, GPS is not suitable for indoor positioning. Furthermore, thermal techniques on Wi-Fi [2], Bluetooth [3], RFID [4], and magnetic fields [5] are capable of realizing superior positioning effects in indoor environments, promoting the application of a wide number of indoor positioning systems (IPS) [6]. Wi-Fi-based IPS, in particular, has become one of the most practicable approaches because of the widespread availability and ease of deployment of Wi-Fi infrastructure [7].

The IPS based on Wi-Fi can generally be divided into two categories [8]: trilateral measurement algorithms and fingerprint-based location algorithms. The trilateral measurement algorithm calculates the distance between the target and the access point (AP) through time of arrival (TOA), time difference of arrival (TDOA), angle of arrival (AOA), and radio wave propagation model (RPM). Such schemes are strongly dependent on complex transmitters and receivers [9], making them difficult to implement on every Wi-Fi

device. In contrast, a fingerprint-based location method is a classic scene analysis algorithm with broad applicability that does not require precise access point position or additional investment in infrastructure and line-of-sight measurement.

A positioning system based on fingerprints is often composed of two stages [10]: the offline training stage and the online positioning stage. In the training stage, some points with known locations are first selected as training points, and then RSSI data are collected from APs detected at each training point. Therefore, the fingerprint of each training point is made up of these RSSI vectors. In general, we always train the RSSI vectors, and the trained vectors are utilized as RP fingerprints. All fingerprints are stored in a database for online localization. In the online stage, RSSI vectors are collected at the corresponding location and transferred to the back-end server. Subsequently, the back-end server matches the received online RSSI vectors with the stored fingerprints to obtain a set of RPs with fingerprints close to the online received RSSI vectors, thereby estimating the target. The fingerprint database is the key to the RSSI-based method. However, in complex indoor scenes, various noise characteristics such as interference, reflection, and refraction can affect signal transmission, resulting in spike noise during RSSI signal acquisition. This actually cannot fully ensure the accuracy and authenticity of data collection in the fingerprint database.

By reason of the foregoing, to improve the accuracy of the fingerprint database and online RSSI vectors with noise reduction, we use JRPCA to train offline fingerprints and online RSSI vectors. The online matching stage often requires traversing the entire fingerprint database, which leads to resource consumption, as well as matching some more distant reference points, increasing the localization error. An efficient clustering strategy is proposed, which divides offline fingerprint data into multiple clusters, and further uses the localization algorithm to find the cluster where the target is located. Finally, the final location is estimated for the user within the potential clusters.

This article is an extension of our conference paper accepted by ICDH2022. In our previous work, the HCS-based clustering method was used to solve the problem of search overhead; however, it did not consider the problem of the existence of boundary point localization in real scenarios. In addition, we also discuss and analyze the proposed method in more detail in this paper and add more test experiments based on real scenarios, aiming to indicate the superiority of the proposed method.

The main contributions of this paper can be summarized as follows:

- The JRPCA model is proposed, which enhances the low-rankness of the fingerprint database using the prior knowledge of singular values to make the data more accurate and thus improve the localization accuracy;
- An effective fingerprint clustering strategy is proposed to reduce the search overhead and radio map size by integrating similar RSSI patterns. A reasonable subset of RPs is obtained adaptively on the basis of predefined RPs to further increase the differences between similar fingerprints.

The rest of this article is organized as follows. We review the related work in Section 2. Section 3 gives an overview of the system and describes the associated processes. Section 4 details the algorithms used in this paper, the JRPCA model optimization algorithm, the one-way hierarchical clustering algorithm, and the localization algorithm. The experiments and the corresponding experimental results are introduced in Section 5. Finally, Section 6 summarizes the whole paper.

## 2. Related Work

This section displays a brief introduction of some relevant studies on indoor location based on the fingerprint. Bahl and Padmanabhan [11] created the first WLAN-based indoor localization system, Radar, and adopted the Euclidean distance to select a few nearby RPs to estimate the location of target points. It should be noted that random noise was not taken into account in their work. Horus employed the probability distribution histogram of each RP in the offline training phase for further position prediction based on the signal's probability distribution [12]. Chai and Yang [13] developed a relatively coarse calibration-

based technique for estimating signal intensity using an interpolation function. Some studies focus on fingerprint training to enhance accuracy. Youssef et al. [14] developed a MaxMean approach to construct the fingerprint database, which helps to pick several strongest APs in the offline stage that can cover the full localization region. Despite the ability of such a strategy in intensifying the robustness of an offline fingerprint database, it excludes unusual occurrences during the online phase.

On the other hand, a variety of clustering techniques have been put out to address the issue of the huge demand for fingerprint storage. One of these is the k-means clustering method [15], which divides the entire wireless map into *k* clusters by a recursive approach. The advantage of low calculation cost has advanced its extensive use in fingerprint localization. Such systems based on k-means clustering, however, fall short of providing perfect localization accuracy, as the random selection of initial cluster members or samples increases the risk of incorrect cluster selection. To increase the localization accuracy by overlapping between clusters produced by the k-means algorithm, two types of enhanced clustering technologies have been proposed [15]: the multiple nearest neighbor (MNN) overlap clustering strategy and the Voronoi (VRN)-based overlap clustering strategy. Although both overlapping strategies are superior to the k-means strategy in terms of localization accuracy, the resulted in higher calculation complexity still cannot be ignored, as shown in [16].

Unlike the k-means clustering technique, affinity propagation (AFP) clustering [17] can acquire the ideal cluster head and its related cluster by iteratively transferring two types of information between data points. It has been widely employed in numerous fingerprint systems. AFP clustering does not require a certain number of clusters to be generated, nor does it require a random selection of samples as input. Nevertheless, when applied to datasets with complex structures, the negative Euclidean distance [18] between samples and individual data points, as a measure of similarity, can dramatically impair the effectiveness of such clustering. Another approach for forming training location clusters based on AP virtual locations was proposed in [19] for fingerprint-based indoor localization. However, this clustering technique performs best in indoor conditions without linear limitations. The hierarchical clustering strategy (HCS) proposed by [20] partitions the fingerprint data into a set of non-overlapping clusters. Each cluster contains the training positions that receive the strongest signal from a certain number of APs, which are organized by hierarchical level definitions to form a fixed sequence. Therefore, the number of clusters created by HCS is easily determined based on the number of APs deployed in the localization region and the hierarchy level selected by [20].

Specifically, compared to k-means clustering and the two overlapping strategies shown in [15], the HCS method assigns a unique ID to each formed cluster based on the order of the strongest signals provided by the AP, thus greatly reducing the search overhead and localization errors. On this basis, we propose a two-phase fingerprint-matching algorithm of JRPCA based on clustering, which uses the JRPCA model to further compress the fingerprint database by training the fingerprints through the augmented Lagrange multiplier (ALM) algorithm [21].

### 3. System Framework

We first establish some fundamental symbols to clearly depict the system framework. In the positioning area, suppose there are n APs and N RPs. The APs' location, transmission power, manufacturers, and owners are not needed to be known. The location of $RP_i$ is $l_i = (x_i, y_i)$, and suppose that all APs can be detected at $RP_i$. We measure the RSSI signal multiple times at each RP, and the average of the RSSI signals collected at each reference point is taken as the fingerprint of that point. Suppose that the fingerprint of $RP_i$ is $f_i = \left( rssi_i^1, rssi_i^2, \ldots, rssi_i^j, \ldots, rssi_i^n \right)$, where $rssi_i^j$ is the RSSI collected from $AP_j$. All RPs' locations form the location dataset $L_N = (l_1, l_2, \ldots, l_N)^T$, and all RPs' fingerprints denote $F_N = \left( f_1^T, f_2^T, \ldots, f_N^T \right)^T$. The offline fingerprints database consists of reference point location and its corresponding RSSI value, where the structure is $\langle L_N, F_N \rangle$.

$$F_N = \begin{pmatrix} f_1 \\ f_2 \\ \vdots \\ f_N \end{pmatrix} = \begin{pmatrix} rssi_1^1, rssi_1^2 \dots rssi_1^n \\ rssi_2^1, rssi_2^2 \dots rssi_2^n \\ \vdots \\ rssi_N^1, rssi_N^2 \dots rssi_N^n \end{pmatrix}_{N \times n}. \tag{1}$$

The proposed Wi-Fi fingerprint localization system adopts a robust noise suppression technique and an efficient clustering method for location estimation in two stages. The framework of the localization system is shown in Figure 1. In the offline phase, the sparse peak noise of the fingerprint database is reduced by the JRPCA model after the fingerprints are collected. The denoised fingerprints are then clustered to reduce the subsequent search overhead by integrating similar RSSI patterns. The offline fingerprint database is constructed by denoising and clustering processes for online matching. In the online phase, online fingerprints are constructed by the same denoising process. After processing the offline fingerprint database and online fingerprints, suitable clusters are matched based on the strongest signal received from the AP by the online fingerprints. Finally, the WKNN algorithm is used to estimate the position of the target point in the selected clusters. The detailed design and working methods of the noise suppression technique, clustering strategy, and localization technique of the fingerprint system proposed in this paper are given in Section 4.

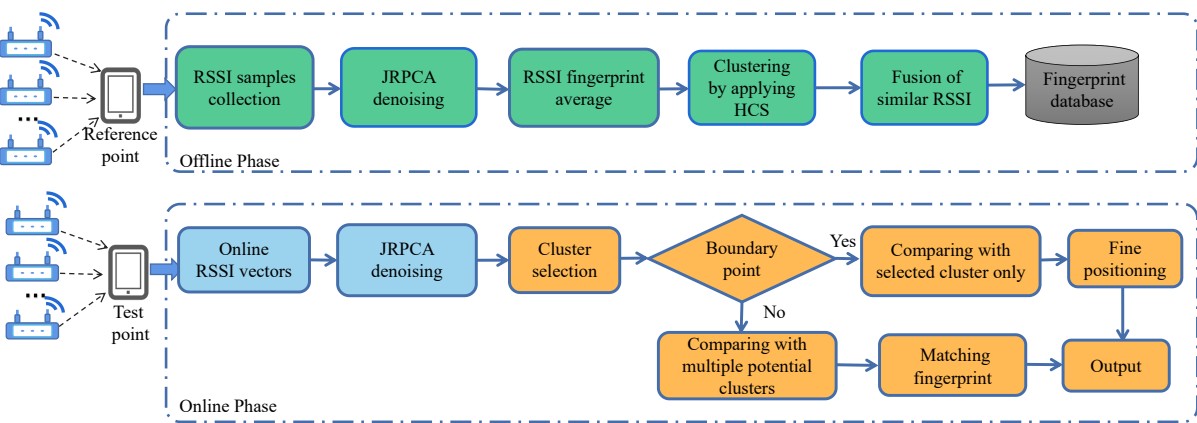

**Figure 1.** Framework of proposed fingerprint positioning system.

## 4. Positioning Algorithm

### 4.1. JRPCA in Offline and Online Phase

In this section, we first introduce the limitations of the RPCA [22] model for training fingerprints. Then, the JRPCA is proposed and the details of solving the JRPCA model by ALM in the offline phase are shown.

#### 4.1.1. RPCA Noise Reduction Optimization Model

Let $F_N$ be the offline fingerprint database constructed by n APs and N RPs, $F_N'$ be the reconstructed database, and $E_N$ be the noise. The RPCA model can be represented as

$$F_N = F_N' + E_N. \tag{2}$$

The problem is to reduce the peak noise $E_N$ and reconstruct robust $F_N'$:

$$\min \operatorname{rank}(F_N') + \gamma \|E_N\|_0 \quad \text{s.t.} \quad F_N' + E_N = F_N, \tag{3}$$

where $\|.\|_0$ is applied to force $E_N$ to be sparse, and the parameter $\gamma$ ($\gamma > 0$) controls the tradeoff between $\operatorname{rank}(F_N')$ and $\|E_N\|_0$.

Due to the non-convex and non-smooth properties of rank and $\|.\|_0$ in optimization, in general, the problem is converted into solving a convex optimization problem. Then (3) is transformed into

$$\min \left\|F_N'\right\|_* + \|E_N\|_1 \quad \text{s.t.} \quad F_N' + E_N = F_N, \tag{4}$$

where $\left\|F_N'\right\|_* + \|E_N\|_1$ is a convex hull of $\text{rank}\left(F_N'\right) + \gamma \|E_N\|_0$ over the set $\left(F_N', E_N\right)$. Therefore, (4) is convex with a unique minimum value.

### 4.1.2. Fingerprint Database Reconstruction Based on JRPCA

Here, we propose JRPCA to improve the fingerprint noise and make the data more accurate, thus improving the positioning accuracy.

The nuclear norm $\|.\|_*$ assigns equal and constant threshold values to all singular values of the matrix, ignoring the different data characteristics represented by different singular values in the matrix. Thus, the concept of weighted is introduced in the nuclear norm optimization model. Relative to the $L_1$-norm, the $L_{2,1}$-norm can produce row (or column) based sparsity, thus improving the model by using the $L_{2,1}$-norm. In this way, the robustness of the model has been enhanced while ensuring no excessive data loss. On this basis, a JRPCA model based on the weighted nuclear norm and $L_{2,1}$-norm is constructed to recover the low-rank matrix of the original data. The model is shown as follows:

$$\min_{F_N', E_N} \left\|F_N'\right\|_{W,*} + \lambda \|E_N\|_{2,1} \quad \text{s.t.} \quad F_N = F_N' + E_N, \tag{5}$$

where $\lambda$ refers to the weight of noise and is a known quantity.

The ALM method is used to solve the JRPCA model in this paper. The solution procedure and the block diagram of the algorithm are described in detail in the next subsection.

### 4.1.3. Model Solution

In this section, the ALM method is adopted to solve the proposed model, which solves the constrained optimization by transforming it into an unconstrained optimization problem. To solve the proposed optimization problem by using ALM, we introduce the preliminary definitions and theorems as follows:

- Shrinkage Operator: For any $\tau > 0$ and $\mathbf{X} \in \mathbb{R}^{m \times n}$, the shrinkage operator $S_\tau(\mathbf{X})$ is defined as

$$S_\tau\left(\mathbf{X}_{ij}\right) = \begin{cases} \mathbf{X}_{ij} - \tau \ x > \tau \\ \mathbf{X}_{ij} + \tau \ x < -\tau \\ 0 \text{ otherwise.} \end{cases} \tag{6}$$

- Soft-thresholding Operator: For any $\tau > 0$ and $\mathbf{X} \in \mathbb{R}^{m \times n}$ with a singular value decomposition $\mathbf{X} = U\Sigma V^T$, the soft-thresholding operator is

$$D_\tau(\mathbf{X}) = U S_\tau(\Sigma) V^T. \tag{7}$$

- For any $\tau > 0$ and $\mathbf{X} \in \mathbb{R}^{m \times n}$, the shrinkage operator is the optimal solution of the function as

$$S_\tau(\mathbf{X}) = \arg \min_{\mathbf{X}} \left\{ \frac{1}{2} \|\mathbf{X} - \mathbf{Y}\|_F^2 + \tau \|\mathbf{X}\|_1 \right\}. \tag{8}$$

- For any $\tau > 0$ and $\mathbf{X} \in \mathbb{R}^{m \times n}$, the soft-thresholding operator is the optimal solution of the function as

$$D_\tau(\mathbf{X}) = \arg \min_{\mathbf{X}} \left\{ \frac{1}{2} \|\mathbf{X} - \mathbf{Y}\|_F^2 + \tau \|\mathbf{X}\|_* \right\}. \tag{9}$$

To solve the optimization problem, we first convert the constrained optimization problem into an unconstrained optimization problem by introducing a Lagrangian multiplier

$Y$ and a quadratic penalty term and then formulating the augmented Lagrange function as follows:

$$L(F'_N, E_N, Y, \mu) = \lambda \|E_N\|_{2,1} + \|F'_N\|_{W,*} + <Y, F_N - F'_N - E_N> + \frac{\mu}{2}\|F_N - F'_N - E_N\|_2^2, \tag{10}$$

where $Y$ is a Lagrange multiplier, $\mu$ is a positive scalar, and $<Y, F_N - F'_N - E_N>$ is an iterative procedure.

The alternating direction method is used to iterate (10), update the matrices $F'_N$ and $E_N$, and loop the algorithm to the termination criterion. The updating process is shown as follows.

1.   Fix $E_N, Y, \mu$, that is, when $E_N = E_k, Y = Y_k, \mu = \mu_k$, iteratively update $F'_N$.

$$\begin{aligned} F'^*_N &= \arg\min_{F'_N} \|F'_N\|_{W,*} + \frac{\mu}{2}\left\|F_N - E_N - F'_N + \frac{1}{\mu}Y\right\|_2^2 \\ &= \arg\min_{F_N} \frac{1}{2}\left\|F'_N - \left(F_N - E_N + \frac{1}{\mu}Y\right)\right\|_2^2 + \frac{1}{\mu}\|F'_N\|_{W,*}. \end{aligned} \tag{11}$$

2.   Fix $F'_N, Y, \mu$, when $F'_N = F'^{\,k}_N, Y = Y_k, \mu = \mu_k$, the matrix $E_N$ in (10) is iteratively updated to obtain:

$$\begin{aligned} E^*_N &= \arg\min_{E_N} \frac{\lambda}{\mu}\|E_N\|_{2,1} + \frac{1}{2}\left\|F_N - E_N - F'_N + \frac{1}{\mu}Y\right\|_2^2 \\ &= \arg\min_{E_N} \frac{1}{2}\left\|E_N - \left(F_N - E_N + \frac{1}{\mu}Y\right)\right\|_2^2 + \frac{\lambda}{\mu}\|E_N\|_{2,1}. \end{aligned} \tag{12}$$

3.   When the matrices $F'_N$ and $E_N$ converge, i.e., $F'_N = F'^{\,k}_N$ and $E_N = E_k$, the matrices $Y$ and $\mu$ in Equation (10) are updated iteratively:

$$\min_Y \frac{\mu}{2}\left\|F_N - F'_N - E_N + \frac{Y}{\mu}\right\|_2^2. \tag{13}$$

The iterative update of $Y$ is obtained from Equation (13) as:

$$Y^* = Y + \mu(F_N - F'_N - E_N), \tag{14}$$

where the iterative update of the positive penalty coefficient $\mu$ is:

$$\mu = \min(\rho\mu, \mu_{\max}). \tag{15}$$

4.   The selection of the weight vector $W(W = [w_1, w_2, \ldots, w_n](w_i \geq 0))$ is the key to the solution. The unknown weight vector $W$ can be obtained by updating the matrix $F_N$. In the matrix, the information of the data represented by the large singular value is more reflective of the important components of the data compared to the small singular value. Therefore, the contraction range of the small singular values can be increased and the contraction range of the large singular values can be decreased to retain the important information in the data. Thus, the singular value $\sigma_i(F'_N)$ $(i = 1, \ldots, n)$ is inversely proportional to the weight vector $W$:

$$w_i = \frac{c\sqrt{n}}{\sigma_i(F'_N) + \tau}, \tag{16}$$

where $c > 0$ is a constant, and $\tau > 0$ ensures that the weights can still be calculated when $\sigma_i(F'_N)$ is 0.

Based on the above discussion, the complete JRPCA algorithm (Algorithm 1) flow is presented here, as shown below:

---

**Algorithm 1:** JRPCA Algorithm

---

 **Intput:** data matrix $F_N \in R^{N \times n}$, parameter $\lambda, \tau$.

 **Initialize:** $F_N'^0 = 0$, $E_N^0 = 0$, $Y_0 = 0$, $\mu_0 > 0$

 **While** $\left\| F_N - F_N' - E_N \right\|_2 > \mu^{-1} \| F_N \|_2$ **do**

  **Calculate the weight**

   $W = [w_1, w_2, \dots, w_n](w_i \geq 0)$

   $w_i = \frac{c\sqrt{n}}{\sigma_i(F_N') + \tau}$

  **when solve**

   $F_N'^{k+1} = \arg\min_{F_N'} L\left(F_N', E_k, Y_k, \mu_k\right)$

  **use**

   $(U, S, V) = \text{SVD}\left(F_N - E_N^k + \mu_k^{-1} Y_k\right)$

   $F_N'^{k+1} = U S_{w/\mu}(S) V^{\mathrm{T}}$

  **when solve**

   $E_N^{k+1} = \arg\min_{E_N} L\left(F_N^{1k+1}, E_N, Y_k, \mu_k\right)$

  **use**

   $E_N^{k+1} = S_{\lambda/\mu}\left(F_N - F_N' + \mu^{-1} Y\right)$

  **Update** $Y_{k+1} = Y_k + \mu_k\left(F_N - F_N'^{k+1} - E_N^{k+1}\right)$

  **Update** $\mu_k$ to $\mu_{k+1}$

  $k \to k+1$

 $F_N' \leftarrow F_N'^k, E_N \leftarrow E_N^k$

 **Output:** $\left(F_N', E_N\right)$

---

*4.2. Proposed Clustering Strategy*

The fingerprint clustering technique proposed in this paper is divided into three main steps. The radio map is first separated into several distinct clusters by the one-way hierarchical clustering strategy's basic operating concept (one-way HCS). The norm is that only the strongest RSSI from a particular AP is sent to the collection of training locations that make up a cluster. Although the RSSI samples measured by a specific AP at a location fluctuate over time, the RSSI values collected by the same AP are spatially correlated. Therefore, the number of clusters created is equal to the number of APs deployed in the location area. Second, the Euclidean distance between any pair of RPs belonging to the same cluster is calculated, and then subsets are generated by fusing RPs whose distance is within a certain range (called threshold). Finally, a representative RSSI vector for each subset in each cluster is calculated. The RPs with similar RSSI in the same subset are fused on average to obtain a new RP representing the subset. Combining our proposed noise reduction algorithm with this clustering strategy, a two-phase fingerprint matching algorithm of JRPCA based on clustering is proposed to further increase the difference between similar RPs and thus improve the localization accuracy.

The whole working process when applying the clustering technique of this paper to the localization region of the 4 APs is shown in Figure 2. Either only one RP or multiple RPs are included in each generated subset. The introduced new parameter threshold (denoted as $\delta$) sets the upper limit of the Euclidean distance between each pair of RSSI-similar reference points that can be classified as a subset of the same cluster. After fusing the RPs of each subset, the fingerprint database construction is completed. The fingerprint of each subset comprises a representative RSSI vector of that subset and the location coordinates of all points in the subset. The RSSI of all reference points of the subset (such as $k$th subset in $i$th cluster) and their position coordinates are averaged and fused separately to obtain the representative RSSI vector. The estimation results of the representative RSSI vector ($r_v$) and location coordinates ($x_v, y_v$) of the subsets are shown below:

$$r_{v_{ik}} = \frac{1}{b} \sum_{j=1}^{b} r^j, \tag{17}$$

where $b$ is the number of reference points within $k$th subset in the cluster $C_i$, and the set of RSSI computed at those b reference points are denoted as $\left[ r^1, r^2, \ldots, r^b \right]$.

$$x_{v_{ik}} = \frac{1}{b} \sum_{j=1}^{b} x_j, y_{v_{ik}} = \frac{1}{b} \sum_{j=1}^{b} y_j, \tag{18}$$

where $[(x_1, y_1), (x_2, y_2), \ldots, (x_b, y_b)]$ denotes the positional coordinates of $b$ training locations within $k$th subset in the cluster $C_i$.

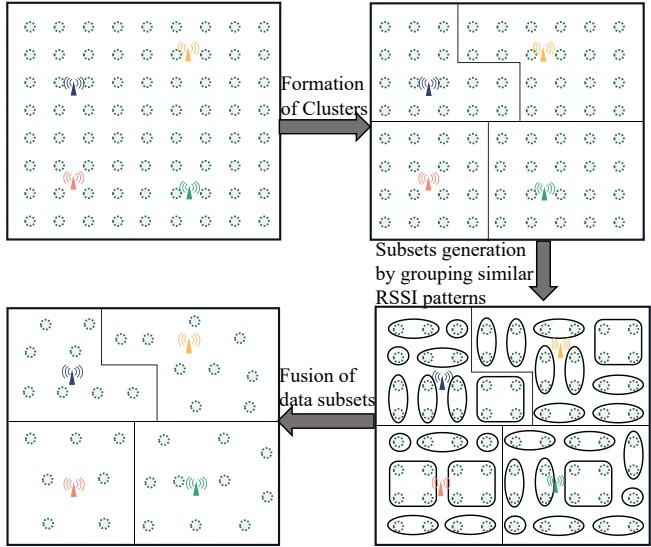

**Figure 2.** The working process of proposed efficient clustering strategy.

As shown in Figure 2, after applying one-way HCS clustering, an important issue is how to choose the initial training location to start the data fusion process within each cluster. In order to divide each cluster into the optimal number of subsets, the strategy we propose uses the RP with the largest RSSI value obtained by AP as the starting point of the data fusion process within the cluster. Formally, for the $i$th cluster ($C_i$), the RP with the largest RSSI obtained from AP$i$ (strongest AP for cluster $C_i$) is used as the initial data point, and then the fusion process starts. Compared with other clustering techniques, this clustering technology greatly reduces the storage requirements of radio maps and the search overhead in the localization phase, and is known as an efficient clustering technique.

Figure 3 details the process of the designed clustering method. In the offline phase, all the pre-defined RPs are divided into clusters, and the cluster with the smallest distance is iteratively integrated into a new cluster. Figure 4 shows the heat map of hierarchical clustering, and the color shades in the figure represent the corresponding RSSI values.

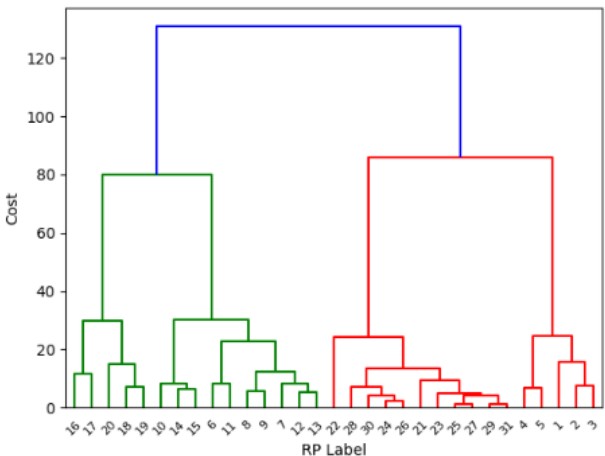

**Figure 3.** Hierarchical clustering process.

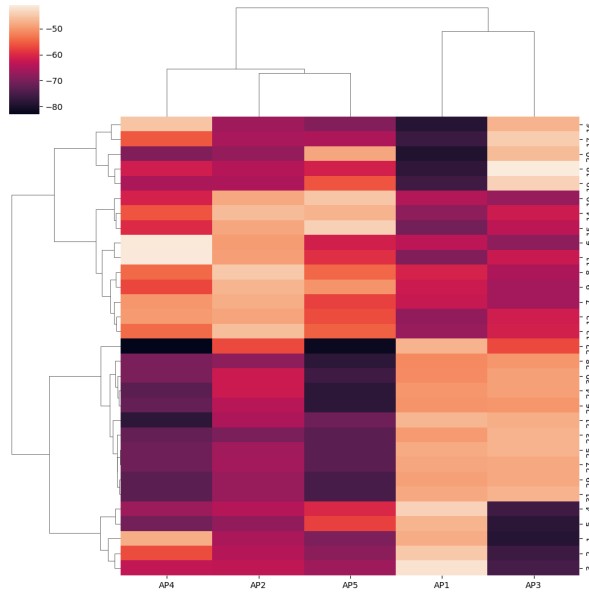

**Figure 4.** Positioning trajectory diagram of different methods.

### 4.3. Online Target Positioning

In this phase, the user terminal receives RSSI information $X$ at the real-time location, and the proposed noise suppression technique is applied to $X$ to obtain the noise-reduced real-time RSSI vector $X'$. Next, the mean value processing is performed to obtain the RSSI information $X''$ of the point to be located. The trained RSSI is then compared to determine the AP with the strongest signal strength, and the potential clusters of $X''$ are identified based on this AP. After mapping a cluster, we filter out the RP of other clusters and use only the RP in the potential cluster region for localization, which greatly reduces the computational effort.

However, in the real scenario, the problem of boundary point localization often arises. At this time, we can no longer directly localize a potential cluster by simply clustering it out, and we must also consider whether other clusters have a greater effect on the target point. Therefore, a boundary point judgment is required.

If the target point $X''$ is a boundary point, $X''$ is divided into multiple neighboring potential clusters and matched with all fingerprint data in these clusters. If the target point $X''$ is not a boundary point, then $X''$ is divided into one potential cluster and matched with all fingerprint data in that cluster only. Finally, the location of the user is further estimated using WKNN based on the location of the known reference points.

WKNN is a popular algorithm improved from the KNN technique [11] with simple computation and high estimation accuracy. The estimation of the WKNN algorithm is based on the Euclidean distance:

$$d_j = \left\| r_j - \hat{r} \right\|_2 \quad \forall j = 1, \ldots, m,$$ (19)

where $\| \cdot \|_2$ is the $\ell_2$-norm operator, $d_j$ is the Euclidean distance, and $r_j$ is the $j$th fingerprint of the potential clusters.

In the WKNN algorithm, the distance values are given weight:

$$w_j = \frac{1}{d_j + \delta},$$ (20)

where $\delta$ is a small positive number introduced in order to control the denominator as not being zero, and $j$ is the index of the reference point that obeys $1 \leq j \leq m$.

Then $k(k > 1)$ reference points with the shortest Euclidean distance are selected as candidate locations in the potential clusters, and the user's location is obtained by averaging the $k$ candidate locations, as follows:

$$(\hat{x}, \hat{y}) = \frac{\sum_{j=1}^{k} w_j (x_j, y_j)}{\sum_{j=1}^{k} w_j}.$$ (21)

## 5. Experiments and Discussion

### 5.1. Experimental Scenarios

We have conducted extensive experiments in laboratories and corridors. The testing area is shown in Figure 5. After analyzing the characteristics of the indoor positioning space, the points are arranged with floor tiles at intervals and collected at each point. The Wi-Fi signal of stable and visible APs will improve the accuracy of fingerprint location results with the increase in the number of selected APs, but it is not infinite. Too many APs will cause mutual interference. Therefore, this paper selects six APs to ensure that the fingerprint reference point in the whole location area can receive Wi-Fi signals to the greatest extent. Then 30 points are randomly selected in the experimental area several times as test points to collect test data. The acquisition time of each reference point and test point is 5 min, and the refresh rate of the sampling equipment is 5 s.

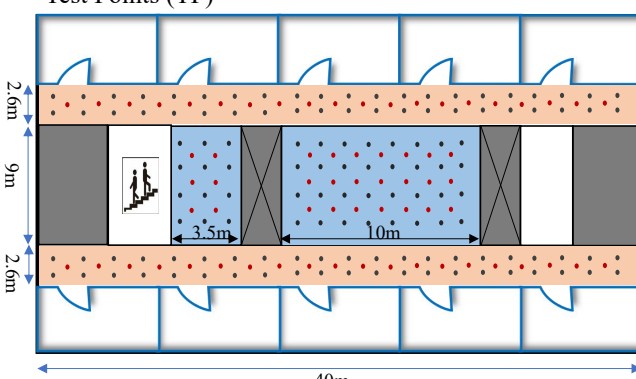

**Figure 5.** Experimental areas.

### 5.2. Analysis of Performance

According to the experimental environment established above, the performance of the algorithm is analyzed.

In this experiment, the weighted *K*-nearest neighbor algorithm (WKNN) is selected as the positioning algorithm, and the *K* values in the WKNN algorithm are 3∼5 for experiments, and the positioning errors of these several *K* values are analyzed. Figure 6 shows that when *K* = 4, most of the positioning error curves of the WKNN algorithm are below the positioning errors of other values. In other words, in the WKNN algorithm, the positioning effect is better when the *K* value is 4. Therefore, the value of *K* will be 4 to carry out the subsequent experiments.

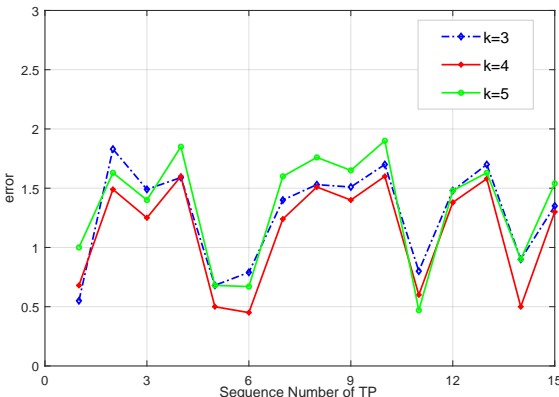

**Figure 6.** Positioning errors of WKNN algorithm under different *K* values.

In our experiments, we compare our method with three advanced schemes: Tilejunction (Tilej.), Radar, and Horus:

- Tilejunction [10]: It maps the target RSSI of each AP to a convex hull termed signal "tile" where the target is likely within. It also partitions the site into multiple clusters to substantially reduce the search space in the LP optimization.
- Radar [11]: It computes the Euclidean distance between the fingerprint and the target RSSI vector, and finds the *k*-nearest neighbors of the smallest distance to estimate the target location.
- Horus [12]: It first calculates the probability distribution of the RSSI value at each RP. Given a target RSSI vector, Horus computes the overall probability of the vector at each RP and finds the one with the maximum likelihood as the target location.

Figure 7 shows the mean localization error versus the number of deployed AP. When the number of AP increases, the localization error decreases because more APs help localize the target to a smaller area. The diminishing returns of adding additional APs are because the signal (or fingerprint) differentiation decreases when we add more APs to a fixed area. Our method achieves the highest accuracy due to the joint consideration of measurement noise and the use of efficient hierarchical clustering. The results show that our method essentially achieves the lowest error compared to other schemes because of the combination of measurement noise considerations and the use of efficient hierarchical clustering.

The fingerprint database is constructed based on original data; PCA algorithm, RPCA algorithm, and JRPCA algorithm are, respectively, used for localization experiments; and the positioning errors of the four fingerprint databases are compared. Figure 8 shows the cumulative distribution function (CDF) of positioning errors in the fingerprint database constructed based on different noise reduction algorithms. The experimental results show that the fingerprint database constructed based on the noise reduction algorithm in this paper is superior to those constructed by the other three algorithms in terms of performance, with 64.2% of the points having a localization error of less than 1 m and 96.8% of the points having a localization error of no more than 2 m.

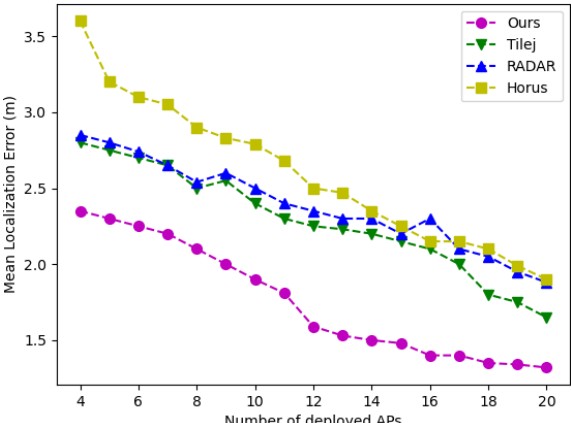

**Figure 7.** Mean error versus the number of AP used.

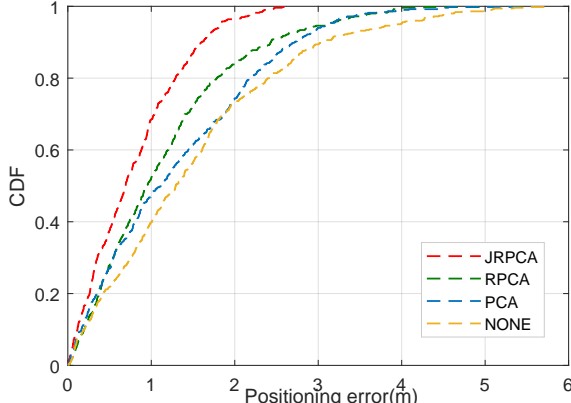

**Figure 8.** CDF of positioning error.

Figure 9 shows the trajectory of the localization results in a 10m×10m hall with the application of the four methods of this paper, Tilej, Radar, and Horus. The blue dots in the figure indicate the actual positions, and the asterisks indicate the estimated positions. During the experiments, initially, the accuracy of these methods was excellent at the localization points, but whenever steering was performed, the error increased, so that the localization accuracy of our method at the steering points was better than the other three methods. As shown in Table 1, among the position estimation of 25 points in the region, the maximum position estimation error of our method is 0.72 m, and the maximum position estimation errors of the other three methods are 1.13 m, 1.34 m, and 1.83 m, respectively. The experimental results show that the performance of the localization technique proposed in this paper is better than the other techniques considered in this paper.

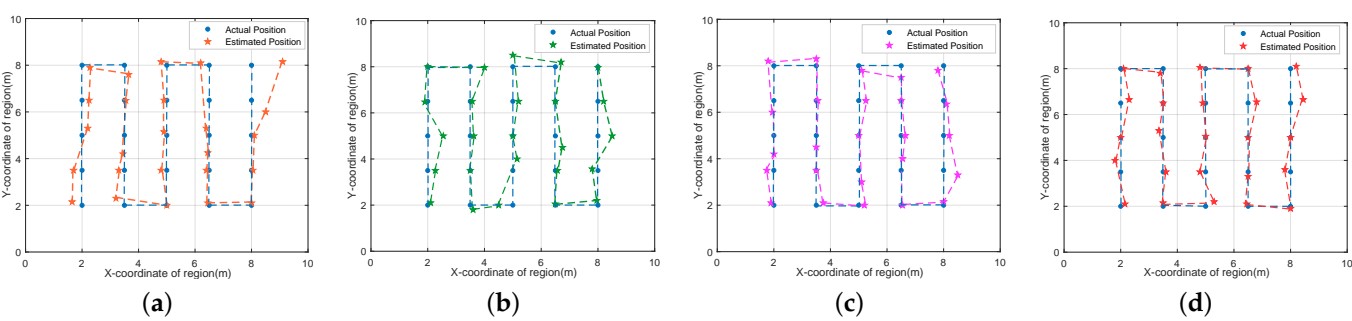

**Figure 9.** Positioning trajectory diagram of different methods: (**a**) Horus; (**b**) Radar; (**c**) Tilej; (**d**) ours.

**Table 1.** Localization error of the four methods.

|                        | Horus | Radar | Tilej | Ours  |
| ---------------------- | ----- | ----- | ----- | ----- |
| Maximum error (m)      | 1.83  | 1.34  | 1.13  | 0.72  |
| Average error (m)      | 0.89  | 0.80  | 0.63  | 0.47  |
| Accumulated error (m)  | 22.25 | 20.75 | 15.75 | 11.75 |

The distribution of localization errors for the four fingerprint techniques based on experiments with real indoor scenes is shown in Figure 10. Due to the complex indoor environment and large measurement noise, the accuracy of Radar is weakened by the scattered nearest neighbors. Horus assumes a certain distribution of signal level at each RP and therefore does not represent the true signal distribution with limited sampling. Therefore, the fingerprint data they collect in complex indoor environments such as lobbies and corridors are inaccurate, resulting in more scattered matching reference points. In contrast, this paper considers the influence of signal noise and adopts a robust noise suppression technique, which makes the RSSI data more accurate and effectively reduces the error in real indoor scenarios.

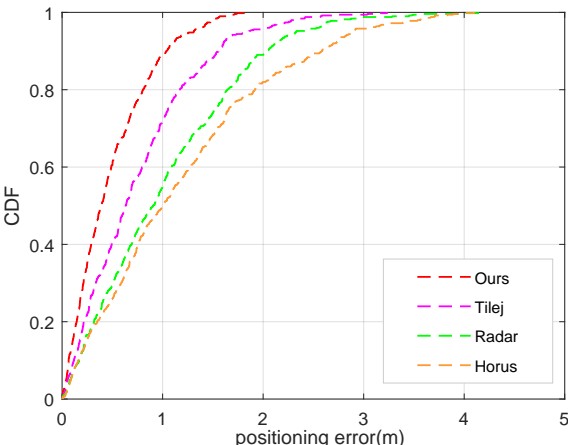

**Figure 10.** Cumulative distribution of localization errors.

## 6. Conclusions and Future Work

Due to the existence of signal measurement noise, the indoor localization method based on fingerprints often results in matching a set of scattered nearest neighbor RPs. Hence, the estimation results are often unsatisfactory. To alleviate this problem and improve robustness, a two-stage fingerprint localization method of JRPCA based on clustering is proposed in this paper. The method estimates target points only in potential clusters, considering the influence of measurement noise during target localization. JRPCA is used to train offline fingerprints and online RSSI vectors. In addition, considering the storage requirement and search cost of radio maps in fingerprint-based indoor positioning systems, a clustering method based on the one-way hierarchy is proposed to obtain reasonable RP clusters adaptively in accordance with predefined RPs. Experimental results demonstrate that the proposed method outperforms other algorithms with respect to robustness and accuracy.

As is well known, WiFi-based indoor positioning technology is easily influenced by different smartphones. This experiment is carried out using only one kind of device without consideration of the influences of different types of devices and receiving terminals on RSSI signals. Therefore, the selection of smart devices is also a significant research topic for indoor positioning. Moreover, the target to be tested is stationary during the experiment, and the localization of the moving target in the WLAN environment is to be solved in the next step. Consequently, future research will further concentrate on exploring the above factors.

**Author Contributions:** Conceptualization, M.Z., Y.X., L.Z., and J.X.; methodology, M.Z. and L.Z.; software, M.Z.; validation, L.Z., M.Z., Y.X., and J.X.; formal analysis, M.Z., L.Z., and J.X.; investigation, M.Z. and Y.X.; resources, L.Z.; data curation, L.Z., M.Z., and Y.X.; writing—original draft preparation, M.Z. and L.Z.; writing—review and editing, M.Z., L.Z., Y.X., and J.X. All authors have read and agreed to the published version of the manuscript.

**Funding:** This work is supported by the National Key Research and Development Program (2018YFB2100301); National Natural Science Foundation of China (61972131).

**Data Availability Statement:** Not applicable.

**Conflicts of Interest:** The authors declare no conflict of interest.

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
