# Peer review of "Cluster-Based JRPCA Algorithm for Wi-Fi Fingerprint Localization"

_electronics, doi:10.3390/electronics12010153_

Round 1

Reviewer 1 Report

This manuscript presents an indoor localization algorithm using the JRPCA based clustering of the nodes. It uses RSSI for clustering. Localization performance is measured in terms on accuracy and robustness. Minor writing style and grammatical improvement need to be done.

Reviewer 2 Report

Due to the existence of signal measurement noise, fingerprint based indoor location methods often lead to matching a group of scattered nearest neighbor fingerprint points, resulting in large location errors. In order to solve this problem, the authors propose a two-stage JRPCA fingerprint matching algorithm based on clustering. Experimental results show that this method is superior to other algorithms in robustness and accuracy. I think there are some major issues in the manuscript that need to be addressed, some of them are listed below.

[1] WiFi fingerprint localization based on clustering has been widely studied, and the processing of locating boundary points has also been widely studied. This paper mainly expands on the basis of the authors' previous conference papers. Although the authors emphasize the differences with the previous conference papers, for example, this paper considers the problem of boundary point localization in real scenarios, the authors do not describe the differences with the existing relevant literature.

[2] Compared with the existing algorithms, the main theoretical contributions of this paper are not clear, in order to highlight the innovation of this paper, it is suggested that the authors emphasize the main contributions of this paper compared with the existing literatures in the introduction.

[3] WiFi fingerprint positioning has been widely used in indoor positioning systems, so it is recommended that the authors add some state-of-the-art indoor positioning algorithms based on WiFi as comparison baselines, which has demonstrated the advantages of this manuscript. The three comparison baseline algorithms selected by the authors were published in 2015, 2000 and 2008, respectively. Therefore, it is recommended that the authors add at least one paper published in recent three years as the comparison baseline.

[4] The depth of experimental analysis is not enough. In Figure 7 to Figure 10, the authors only draw a simple conclusion that the positioning accuracy of the algorithm proposed in this paper is higher than that of the three existing algorithms, and does not analyze the specific reasons.

[5] As we all know, WiFi-based indoor positioning technology is easily affected by different types of smartphones. Has this issue been considered in this manuscript. What is the device used to collect RSSI values in this paper.

[6] In this paper, 30 points are randomly selected as test points in the experimental area. Because different test points will lead to different positioning errors, the experimental results in this paper are obtained under one random selection or multiple random selection.

Reviewer 3 Report

The authors propose an indoor localization algorithm based on fingerprints provided by Wi-Fi data. Using a Joint-norm robust principal component analysis, they improve the localization accuracy via the proposed two-stage matching technique. 

The work at hand is well-written without grammatical issues. Its novelty is remarkable, and the authors' findings are evaluated extensively during the experimental protocol. Therefore, I find the article interesting and worth publication. However, as the authors highlight the computational complexity factor for their method's selection, it would be helpful to provide some experimental results about this aspect of their work.

Round 2

Reviewer 2 Report

The authors have answered my comments, acceptance is suggested